# Dynamic Analysis of the Coracohumeral Ligament Using Ultra-Sonography in Shoulder Contracture

**DOI:** 10.3390/s23084015

**Published:** 2023-04-15

**Authors:** Tatsuo Kato, Atsuyuki Inui, Yutaka Mifune, Hanako Nishimoto, Tomoya Yoshikawa, Issei Shinohara, Takahiro Furukawa, Shuya Tanaka, Masaya Kusunose, Ryosuke Kuroda

**Affiliations:** Department of Orthopaedic Surgery, Kobe University Graduate School of Medicine, 7-5-1 Kusunoki-cho, Chuo-ku, Kobe 650-0017, Japan; t.kato.ort@gmail.com (T.K.); m-ship@kf7.so-net.ne.jp (Y.M.); hanako-nishi@live.jp (H.N.); tomo_yoshi_0926@yahoo.co.jp (T.Y.); 203m878m@stu.kobe-u.ac.jp (I.S.); taka1023@med.kobe-u.ac.jp (T.F.); shuyatanaka0517@gmail.com (S.T.); abcd102317@gmail.com (M.K.); kurodar@med.kobe-u.ac.jp (R.K.)

**Keywords:** CHL, ultrasonography, shoulder contracture, PIV, dynamic analysis

## Abstract

The coracohumeral ligament (CHL) is related to the range of motion of the shoulder joint. The evaluation of the CHL using ultrasonography (US) has been reported on the elastic modulus and thickness of the CHL, but no dynamic evaluation method has been established. We aimed to quantify the movement of the CHL by applying Particle Image Velocimetry (PIV), a technique used in the field of fluid engineering, to cases of shoulder contracture using the US. The subjects were eight patients, with 16 shoulders. The coracoid process was identified from the body surface, and a long-axis US image of the CHL parallel to the subscapularis tendon was drawn. The shoulder joint was moved from 0 degrees of internal/external rotation to 60 degrees of internal rotation at a rhythm of one reciprocation every 2 s. The velocity of the CHL movement was quantified by the PIV method. The mean magnitude velocity of CHL was significantly faster on the healthy side. The maximum magnitude velocity was significantly faster on the healthy side. The results suggest that the PIV method is helpful as a dynamic evaluation method, and in patients with shoulder contracture, the CHL velocity was significantly decreased.

## 1. Introduction

The classic description of the coracohumeral ligament (CHL) originates from the lateral side of the horizontal limb of the coracoid process, inserts into the greater and lesser tuberosities, and covers the rotator interval, the space between the supraspinatus and subscapularis muscles [1]. Other histologic elements present between the coracoid process and the subscapularis muscle include the subcoracoid fat pad, the subcoracoid bursa, and the subscapular recess of the glenohumeral joint [2]. Yang et al. grossly examined the insertion of the CHL in 26 fresh-frozen cadavers. They showed that in three shoulders, the CHL branched and was inserted into both the supraspinatus and subscapularis tendons. In one shoulder, the ligament was inserted only into the subscapularis tendon [3]. They also found that CHL was a thickening of the joint capsule by comparing the CHL base with the intersection of the CHL and the joint capsule and comparing the joint capsule with the coracoacromial ligament. Boardman et al. considered the CHL as an important capsuloligamentous structure of the glenohumeral joint [4]. The CHL acts in combination with the rotator cuff muscles, superior glenohumeral ligament, and capsule to contribute to the stability of the glenohumeral joint. In particular, it functions to prevent inferior subluxation of the humeral head and plays an important role in stabilizing the glenohumeral joint in the upright position [5]. Arai et al. examined the structure between the CHL and subscapularis histo-anatomically in nineteen cadavers and discussed the function of the ligament. The CHL encompasses the entire subscapularis muscle and its insertion, indicating that it functions as a holder for the subscapularis and supraspinatus muscles. Furthermore, the ligament is composed of irregular and sparse fibers and contains relatively high amounts of type III collagen, indicating that it is flexible [6]. Frozen shoulder, also known as adhesive capsulitis (ACS), is a condition in which the shoulder joint becomes stiff, painful, and difficult to move. It is a relatively common condition, affecting approximately 2% to 5% of the general population, with a higher incidence in women and individuals over the age of 40 [7]. The cause of ACS is not entirely clear, but it is believed to occur when the capsule that surrounds the shoulder joint becomes thick and tight, leading to a decrease in the joint’s range of motion. Regarding CHL and the shoulder joint range of motion (ROM), there are several reports, including a report indicating that contracture of the CHL and rotator interval is the main cause of chronic ACS [8], a report showing a correlation between CHL thickness and abduction and external rotation ROM in ACS [9], and a report indicating that the shoulder joint range of motion is worse in middle-aged and older adults as the elastic modulus of the CHL increases [10]. In these reports, CHL has been evaluated statically by ultrasonography (US) but not dynamically. In recent years, Particle Image Velocimetry (PIV), a fluid engineering technique that can visualize fluid velocity, has been attracting attention in the field of medicine as a method for dynamic analysis, and it is used to analyze blood flow in cerebral aneurysms and to evaluate airflow in the vocal cords [11,12]. Using the PIV method in the orthopedic field, a report assessed the relationship between fascial gliding and postoperative pain after proximal femoral fracture surgery [13]. In this report, postoperative pain significantly increased as the gliding of the vastus lateralis muscle decreased. Shinohara et al. reported that the PIV technique was applied to patients with triangular fibrocartilage complex (TFCC) injuries, where an increase in TFCC velocity was observed during wrist motion in patients with TFCC injuries [14]. Tanaka et al. reported that the vertical velocity of the extensor carpi ulnaris (ECU) tendon during ulnar deviation was higher in the TFCC injury group by using the PIV method [15]. Thus, observations of dynamic changes in the TFCC and ECU tendon by the PIV method may lead to the diagnosis of TFCC injury. Magnetic resonance imaging (MRI) and computed tomographic arthrography (CTA) are commonly used to diagnose TFCC injury, but these imaging modalities are costly and radiation-exposed, whereas US has the advantages of a low cost and no radiation exposure. Kim YS et al. reported a strain analysis of intact supraspinatus tendons by US with PIV. They found different strains in the superficial and deep layers of intact live supraspinatus tendons and that the pattern of strain and displacement varied in response to isometric and isotonic shoulder joint motion [16]. They believe that the results of this shoulder motion analysis suggest that the avulsed rotator cuff tear must be repaired layer by layer separately to withstand the uneven strain after repair. Thus, it was shown that the use of the PIV method for US movies allows for the dynamic evaluation of tendons and ligaments. In this study, we hypothesized that CHL mobility is reduced in a shoulder contracture patient, and CHL mobility was evaluated using ultrasound and PIV.

## 2. Materials and Methods

### 2.1. Population

The subjects who had shoulder contracture were evaluated in the study. Shoulder contracture was defined as the following: the range of motion was less than 120 degrees in anterior elevation or less than 30 degrees in external rotation in the drooping position [17]. There were eight cases with sixteen shoulders (eight on the affected side, eight on the healthy side), and their mean age was 59.4 years old. In those cases, the average preoperative range of motion was 94.3 degrees of anterior elevation, 81.4 degrees of abduction, and 42.9 degrees of external rotation. Using G*Power 3.1, a power analysis was conducted based on data from the pilot study to determine the sample size. Previous sample size calculations indicated that a sample size of eight participants would be sufficient to detect a difference in velocity of 1.4 mm/s in the affected and healthy sides using a t-test (with an effect size of 1.02, α = 0.05, and power of 0.8).

### 2.2. Experimental Data Acquisition and Analysis

The coracoid process was identified from the body surface, and the 15 MHz linear probe (Canon Aplio 300, TUS-A300, Canon Medical Systems, Tochigi, Japan) was placed parallel to the subscapularis tendon to delineate the long-axis image of the CHL (Figure 1). The triangular-in-shape hyperechoic structure located between the subscapularis tendon, the coracoid process, and the deltoid muscle was defined as the CHL [10] in the present study and set as the region of interest (Figure 2). For motion analysis, active internal or external rotation of the shoulder joint was performed in 0 degrees of abduction and 0 degrees of anterior elevation position. First, the shoulder was moved from 0 to 60 degrees of internal rotation at a rhythm of one round trip (N to IR to N position) every 2 s, paced by a metronome (Figure 1). The same examination was performed five times per shoulder. Three examiners evaluated all US images. The velocity magnitude of the CHL motion was analyzed by using PIV fluid measurement software (PIV lab. Version 2.36, add-in software from MATLAB (Mathworks, Natick, MA, USA)) (see Appendix A). We edited all US movies into 30 static images per second. The pixel displacement between two sequential images was measured, and the velocity magnitude of the structure inside the region of interest was subsequently calculated. The average and maximum velocities of the CHL were measured. The movement speeds at internal rotation angles from 0 to 20 degrees, from 20 to 40 degrees, and from 40 to 60 degrees during internal and external rotation were also measured (Figure 3).

We also evaluated the correlation between the velocity of CHL and the thickness of CHL. Since the thickness of CHL is not uniform, we measured the CHL thickness just lateral to the coracoid process based on a previous report [18]. The thickness was calculated using ImageJ software (USA National Institutes of Health (NIH), Bethesda, MA, USA), a public-domain Java-based image-processing software developed at the United States NIH. Then, the thickness of the CHL at 0 degrees of internal rotation and 60 degrees of internal rotation was measured, and their mean values were evaluated in correlation with the mean velocity and the maximum velocity of CHL.

To confirm reproducibility, we calculated the intra-rater and inter-rater correlation coefficients using these measurement results. PIV measurements were performed twice, with three examiners performing the measurements. The t-test was used for a magnitude velocity comparison. The relationship between the velocity of CHL and the thickness of CHL was evaluated by using Pearson’s correlation coefficient. The intraclass correlation coefficient (ICC) was calculated by R studio (version 3.6.1; R studio, Boston, MA, USA). The Ethics Committee of our institute approved this study (No. B21009), and informed consent was obtained from all patients involved.

## 3. Results

The mean velocity of CHL was 1.43 ± 0.08 mm/s on the healthy side and 1.27 ± 0.08 mm/s on the affected side (*p* < 0.05), significantly faster on the healthy side. The maximum velocity was 5.88 ± 0.05 mm/s on the healthy side and 4.42 ± 0.02 mm/s on the affected side (*p* < 0.05), significantly faster on the healthy side (Figure 4). Regarding the velocity of each range of motion during internal and external rotation, the data for the affected and healthy sides are shown in Figure 5. During the internal rotation, the CHL contracts and moves toward the coracoid process, and during external rotation, the CHL extends and moves toward the humerus (Figure 6). During internal rotation at 0 to 20 degrees, the mean velocity of CHL was 1.45 ± 0.12 mm/s on the healthy side and 1.24 ± 0.1 mm/s on the affected side (*p* = 0.09). During internal rotation at 20 to 40 degrees, the mean velocity of CHL was 2.06 ± 0.18 mm/s on the healthy side and 1.43 ± 0.11 mm/s on the affected side (*p* = 0.005), significantly faster on the healthy side. During internal rotation at 40 to 60 degrees, the mean velocity of CHL was 1.28 ± 0.14 mm/s on the healthy side and 1.25 ± 0.1 mm/s on the affected side (*p* = 0.83). During external rotation at 60 to 40 degrees, the mean velocity of CHL was 1.89 ± 0.16 mm/s on the healthy side and 1.3 ± 0.1 mm/s on the affected side (*p* = 0.003), significantly faster on the healthy side. During external rotation at 40 to 20 degrees, the mean velocity of CHL was 1.83 ± 0.16 mm/s on the healthy side and 1.36 ± 0.11 mm/s on the affected side (*p* = 0.02), significantly faster on the healthy side. During external rotation at 20 to 0 degrees, the mean velocity of CHL was 1.01 ± 0.09 mm/s on the healthy side and 0.94 ± 0.09 mm/s on the affected side (*p* = 0.53). The inter-rater reliability was 0.79, and the intra-rater reliability was 0.80 for this assessment.

The mean CHL thickness was 1.65 ± 0.15 mm on the affected side and 1.21 ± 0.08 mm on the healthy side (*p* = 0.018). The mean CHL thickness was correlated with the mean velocity of CHL (correlation coefficient, −0.75; *p* = 0.043) and the maximum velocity of CHL (correlation coefficient, −0.84; *p* = 0.019).

## 4. Discussion

Several reports have noted a correlation between CHL and the shoulder range of motion (ROM). Kanazawa et al. reported that thickening of the CHL correlates with ROM limitations in forward elevation and external rotation based on MRI evaluation [19]. They also reported that the elastic modulus of the CHL is negatively correlated with forward flexion, external rotation (ER) in a neutral position, and 90° abduction with ER in the 30° ER position and tended to increase with age, irrespective of the side of dominance [10]. Physical therapy is generally the first choice of treatment for shoulder contractures, but surgical treatment may be necessary if ROM does not improve, and there have been several reports targeting CHL in the surgical treatment of shoulder contractures. The release of the CHL from the coracoid process with arthroscopic rotator cuff repair was effective in restoring the ROM of ER in small to middle rotator cuff tears at 3 months postoperatively [20]. Thickening of the CHL was observed in ACS, and US-guided release by a Beaver Mini Blade improved the ROM for ER limitation due to ACS. Passive ER with the arm by the side significantly increased from an average of 18 degrees preoperatively to 47 degrees immediately after CHL release [21]. Koide et al. reported that the arthroscopic release of thickened CHL improved the restriction of internal rotation (IR). They consider that the thickening of the CHL at the rotator interval has been thought to be related to ER, but the limitation of IR is related to the thickened CHL from the coracoid base to the superomedial capsule [22]. Many reports pointed out the correlation between CHL thickness and ROM using a static image of the US [18,21,23]. Do et al. measured the thickness of CHL during shoulder ER. Their results showed that the thickness of the CHL was significantly greater in the ACS shoulder (affected shoulder: 2.7 ± 0.6 mm; unaffected shoulder: 1.5 ± 0.4 mm) [18]. In our study, CHL thickness was evaluated as a static assessment. Similar to previous reports, CHL was significantly thicker on the affected side (affected side: 1.65 ± 0.15 mm; healthy side 1.21 ± 0.08 mm (*p* = 0.018)). Our results are smaller than those reported by Do et al. This may be due to the fact that Do et al. measured CHL thickness at the maximum ER position, whereas we measured at the neutral position. We further evaluated the dynamics of CHL using US movies and PIV methods. The results from the present study showed a negative correlation between CHL thickness and the velocity of CHL (the mean velocity of CHL: correlation coefficient, −0.75; *p* = 0.043, the maximum velocity of CHL: correlation coefficient, −0.84; *p* = 0.019). This indicates that the thicker the CHL, the slower its movement. These findings suggest that CHL thickness and its dynamics may be important factors to consider in the assessment and treatment of shoulder contracture.

One such drawback that needs to be taken into account while conducting the US study is the difficulty associated with tracking the object due to anisotropy. In this context, it is important to note that anisotropy refers to the directional dependence of a physical property which can vary depending on the angle of incidence. This makes it challenging to accurately track the movement of the object under study. To overcome this limitation, the cross-correlation method using Particle Image Velocimetry (PIV) was used in the US study. PIV is known to be relatively robust against loss due to anisotropy since it considers the bright spots in the US image as particles, tracks their movement in the region of interest for each frame, and reflects the average of the movement of particles. This method allows for accurate tracking of the movement of the object of interest and helps to mitigate the effects of anisotropy. For that reason, the PIV method has been used in various studies to measure dynamic analysis. For example, Kawanishi et al. reported on the sliding of the vastus lateralis muscle and subcutaneous tissue after proximal femur fracture surgery using the PIV method [13]. The results showed high reproducibility with intra-rater reliability of 0.92 and inter-rater reliability of 0.83. Similarly, Shinohara et al. used the PIV method to measure distal radioulnar joint movements, and their results showed intra-rater reliability of 0.97 and inter-rater reliability of 0.89, indicating high reproducibility [14]. Therefore, the dynamic analysis using the US has been established with high reliability, as demonstrated by the above studies. In our study, the reproducibility of this method on the shoulder joint was found to be lower than that reported above. This is partly due to the greater motion of the shoulder joint compared with the wrist joint, which makes it challenging to accurately track the movement of the shoulder joint using the PIV method. However, the inter- and intra-rater reliability were 0.79 and 0.80, respectively, in this study, indicating that the PIV method is a reliable method for the dynamic assessment using US, even for the shoulder joint.

In our study, we evaluated the velocity of the CHL from 0 to 60 degrees of IR in 0 degrees of abduction and 0 degrees of anterior elevation position. According to a previous report on the strain measurement of CHL in cadavers, statistically significant positive strains were obtained at ER with the arm at 0° elevation, and no positive strain on CHL was observed at 90° abduction with ER or during flexion with ER [5]. We therefore evaluated CHL during IR and ER at 0 degrees of abduction and 0 degrees of anterior elevation position. In previous reports of US evaluation of the CHL thickness and elastic modulus, the US examination was performed in ER in the supinated or abducted position [18,21,23], but such limb positions are often difficult in patients with strong contractures because many cases of shoulder contractures have pain during ER. On the other hand, a few patients with ACS have difficulty with IR in the drooping position. Koide et al. reported that the restriction of IR in the neutral position is almost absent in patients with ACS, even though the restriction of other IR such as hand behind back, horizontal flexion, and IR in flexion and abduction are present [22]. Actually, all cases in our study were able to perform IR in the neutral position without problems. Kimura H et al. reported that US-guided fascia hydrorelease on the CHL in patients with shoulder contracture improved the ROM not only in ER but also in flexion, extension, abduction, and IR [24]. The anterior part of the CHL envelopes the subscapularis muscle, and the posterior part envelopes the supraspinatus/infraspinatus muscle, anchoring the muscles to the coracoid process. Treatment for CHL might affect the muscle condition connected to CHL and improve the ROM in various aspects. Therefore, we believe that the analysis of the dynamics of the CHL simply in IR movements can provide some information on the shoulder with flexion and abduction contractures. During IR, the velocity was maximal at 20–40° on both the healthy and affected sides, with a significant difference between the healthy and affected sides at 20–40°. During ER, the velocity was maximal at 60–40° on the healthy side, while it was maximal at 40–20° on the affected side. This suggests that the tension on the CHL may be applied differently during IR and ER. During IR, the tension on the CHL is thought to be reduced, and in this study, the shoulder motion stopped once at 0° and 60°, suggesting that the velocity of CHL was maximal at 20–40°. On the other hand, during ER, the CHL is thought to be under tension, and in this study, on the healthy side, the CHL moves fastest at 60–40°, where ER begins, suggesting that the CHL is most tensioned at the beginning of ER. On the affected side, it is possible that there is not enough tension on the CHL at the beginning of ER such that the CHL movement at 60–40° was slower.

This study has several limitations. First, only shoulder contracture with a rotator cuff tear was observed, and the evaluation was not performed on a contracture without a rotator cuff tear. The pathophysiology of shoulder contractures due to ACS and those due to rotator cuff tears might differ. The possibility cannot be ruled out that rotator cuff tears without contractures might also result in a decreased velocity of CHL movement. Second, patients who underwent both surgical treatment for rotator cuff tears and manipulation for shoulder contracture were included in this study; however, a postoperative assessment was not performed. It would be interesting to evaluate the CHL velocity after surgical treatment to further understand the pathophysiology of shoulder contractures. Finally, dynamic evaluation by the cross-correlation method provides relatively high reproducibility, but anisotropy and technical reproducibility issues remain with US images. We hope this method will be widely used in clinical practice and further develop the elucidation of pathological conditions using US imaging.

## 5. Conclusions

The observation of CHL by using US images for shoulder contracture has been reported. We performed dynamic analyses of the CHL to compare the affected and healthy sides of patients with shoulder contracture. The velocity of CHL was significantly decreased (11.2% in mean velocity and 24.8% in maximum velocity in this study under that condition) on the affected side. The mean CHL thickness was significantly increased on the affected side and had a negative correlation with the mean velocity of CHL and the maximum velocity of CHL. An analysis of the dynamics of CHL may lead to a better understanding of the pathophysiology of shoulder contractures and the appropriate treatment of them.

## Figures and Tables

**Figure 1 sensors-23-04015-f001:**
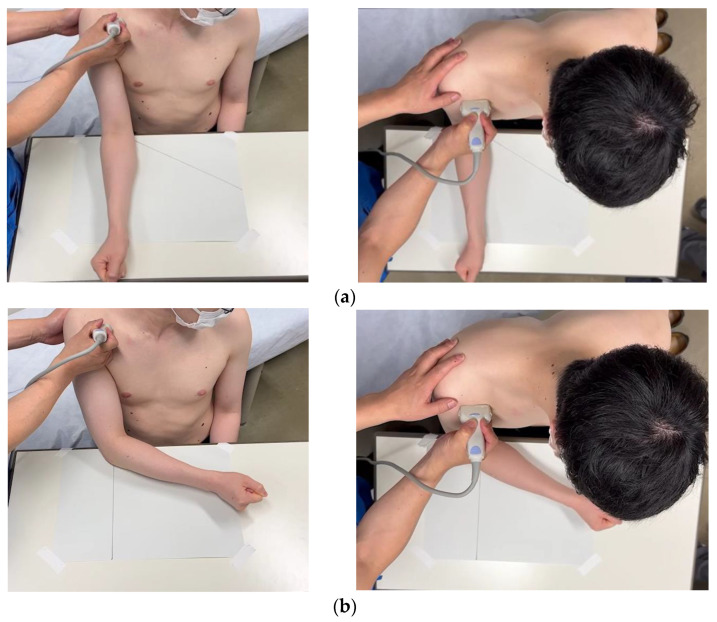
The coracoid process was identified from the body surface, and the CHL’s long-axis image was drawn parallel to the subscapularis tendon. Active shoulder joint movement in the spontaneous drooping position was performed from 0 to 60 degrees of internal rotation. (**a**) Position N (0 degrees of internal rotation) and (**b**) position IR (60 degrees of internal rotation).

**Figure 2 sensors-23-04015-f002:**
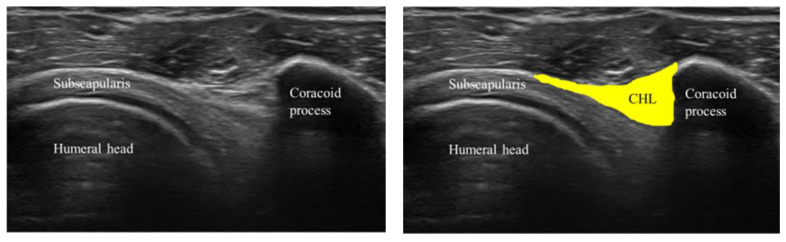
The triangular-in-shape hyperechoic structure located between the subscapularis tendon, the coracoid process, and the deltoid muscle was defined as the CHL in the present study.

**Figure 3 sensors-23-04015-f003:**
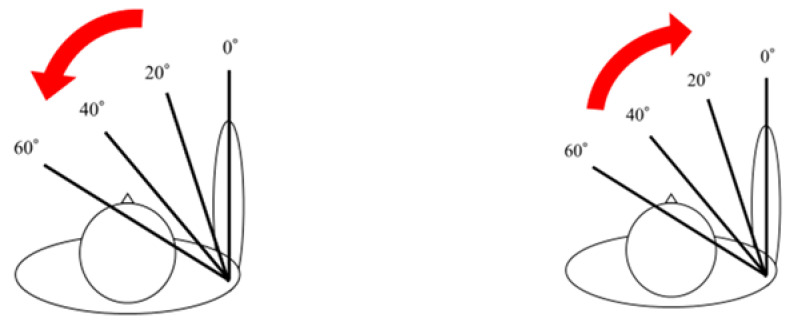
The movement speeds for each angle of 0–20, 20–40, and 40–60 degrees during internal and external rotation were measured.

**Figure 4 sensors-23-04015-f004:**
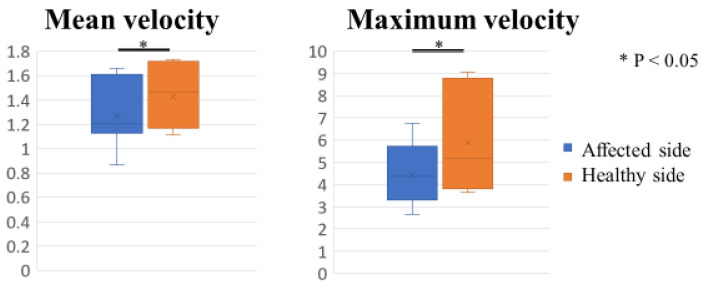
The mean velocity of CHL was significantly faster on the healthy side, and the maximum velocity was significantly faster on the healthy side.

**Figure 5 sensors-23-04015-f005:**
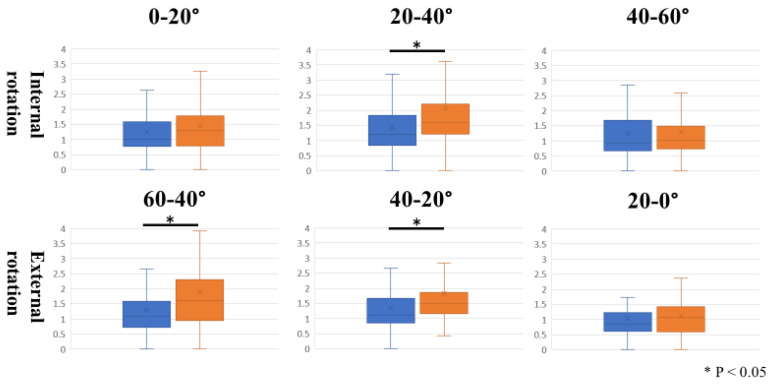
The affected side was significantly slower during internal rotation at 20 to 40 degrees, and the affected side was significantly slower during external rotation at 60 to 40 degrees of internal rotation and 40 to 20 degrees of internal rotation.

**Figure 6 sensors-23-04015-f006:**
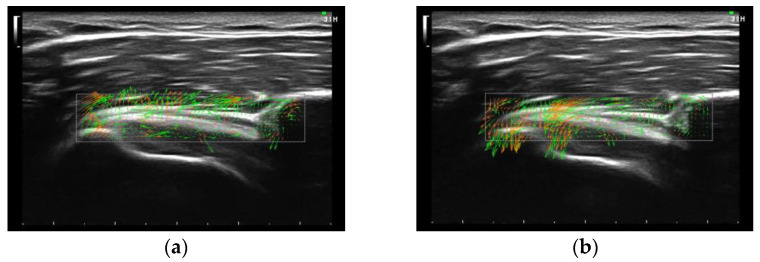
Flow PIV fluid measurement software of CHL movement (**a**) in internal rotation and (**b**) in external rotation. The direction of the arrow indicates the direction in which the particle moves.

## Data Availability

The data presented in this study are available upon request from the corresponding author. The data are not publicly available because of confidentiality concerns.

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
