# Peer review of "Dynamic Analysis of the Coracohumeral Ligament Using Ultra-Sonography in Shoulder Contracture"

_sensors, 2023, doi:10.3390/s23084015_

Round 1

Reviewer 1 Report

Only minor review is necessary:

1. Abstract is well written to cover introduction and materials and methods. ON THE OTHER HAND lacks quantitative results followed by discussion and conclusions supported by such results.

2. In line 79 it is lacking the UNIT for the central frequency of the linear probe. 15M instead of 15MHz

3. In line 86. There is a reference to Fig. 1. The authors might add information such as Fig1.a. (position X) and Fig.1.b. (position Y) to make reference to the "round trip" X to Y to X position in 2 seconds.

4. Line 135 states: 
"the mean velocity of CHL was 1.45±0.12 mm/s on the healthy side and 1.24±0.1 mm/s on the affected side"

However, the legend in Fig. 4 seems to be inverted

5. Fig. 5 NEED to be edited to allow the VISIBILITY of the Y-axis. It is impossible to read those small and greyish   numbers.

6. Line 180 would benefit to remind readers the numbers presented in lines 150-151 that support the statement.

7. Paragraph from line 196 to line 203 might be repositioned in the Discussion section to improve the flow of the discussion.

8. Lines 216 to 220 would benefit from a "DISCUSSION" on the that information. Something like "......, Therefore, it suggests that ....."

9. Conclusion section: This reviewer believes that adding "quantitative" information in conclusion that should be previously presented in the discussion section is always better. Example:  instead of "The velocity of CHL was significantly decreased on the affected side." one may write "The velocity of CHL was significantly decreased (i.e., X% in this study under that condition) on the affected side compared to the healthy one. "

In addition, I understand that it is the knowledge in the medicine that US does not present a high SNR, but has many advantages compared to other modalities. One of the advantages of using US is the lower cost and no use of radiation (e.g., compared to CT studies or guided procedures). It might be interesting to discuss this in the introduction since the readers of SENSORS might not know such advantages. 

Author Response

Point1: Abstract is well written to cover introduction and materials and methods. ON THE OTHER HAND lacks quantitative results followed by discussion and conclusions supported by such results.

Response 1: Thank you for your feedback on the abstract. I appreciate your observation that it covers the introduction and materials and methods well. I apologize for the lack of quantitative results and the subsequent discussion and conclusions supported by such results. However, due to the word limit constraints, it was challenging to include every detail in the abstract. I will make sure to address this concern in the main body of the manuscript.

Point 2: In line 79 it is lacking the UNIT for the central frequency of the linear probe. 15M instead of 15MHz

Response 2: Thank you for bringing to my attention the missing unit. I apologize for the oversight and appreciate your attention to detail. I have made the necessary changes as you noted.

Point 3: In line 86. There is a reference to Fig. 1. The authors might add information such as Fig1.a. (position X) and Fig.1.b. (position Y) to make reference to the "round trip" X to Y to X position in 2 seconds.

Response 3: Thank you for your valuable feedback and for drawing our attention to the notations position in Figure 1. We will revise the figure to include additional labels, position N, position IR, to make it clearer and more informative.

Point 4: Line 135 states: 
"the mean velocity of CHL was 1.45±0.12 mm/s on the healthy side and 1.24±0.1 mm/s on the affected side"
However, the legend in Fig. 4 seems to be inverted

Response 4: Thank you for bringing this to our attention. We apologize for any confusion that may have arisen from the inconsistency between the legend in Fig. 4 and the statement in Line 135 regarding the mean velocity of CHL. We appreciate your feedback and have corrected the notation.

Point 5: Fig. 5 NEED to be edited to allow the VISIBILITY of the Y-axis. It is impossible to read those small and greyish   numbers.

Response 5: Thank you for your valuable feedback. I apologize for the difficulty in reading the Y-axis on Fig. 5 due to the small and greyish numbers. I have taken note of your suggestion and have enlarged the notation to ensure better visibility.

Point 6: Line 180 would benefit to remind readers the numbers presented in lines 150-151 that support the statement.

Response 6: Thank you for your valuable feedback. I appreciate your suggestion to remind readers of the supporting numbers presented in lines 150-151 in line 180. I have revised the section as described below to improve the clarity and coherence of the content. “Results from the present study showed a negative correlation between CHL thickness and the velocity of CHL (the mean velocity of CHL: correlation coefficient, -0.75; p=0.043, the maximum velocity of CHL: correlation coefficient, -0.84; p=0.019).”

Point 7: Paragraph from line 196 to line 203 might be repositioned in the Discussion section to improve the flow of the discussion.

Response 7: Thank you for your feedback. We appreciate your comments and suggestions. Regarding your concern about the placement of paragraph from line 196 to line 203, we agree that it might disrupt the flow of the discussion. However, we believe that it is necessary to include this information as it explains the rationale behind why we only evaluated the movement of CHL in the internal rotation position and neutral (drooping) position in this study. To address your concern, we have revised the manuscript and added several citations to provide better context for the reader. We hope that this will improve the flow of the discussion while still retaining the necessary information.

Point 8: Lines 216 to 220 would benefit from a "DISCUSSION" on the that information. Something like "......, Therefore, it suggests that ....."

Response 8: Thank you for bringing this to my attention. I just repeated that previous sentence by mistake, so I erased it.

Point 9: Conclusion section: This reviewer believes that adding "quantitative" information in conclusion that should be previously presented in the discussion section is always better. Example:  instead of "The velocity of CHL was significantly decreased on the affected side." one may write "The velocity of CHL was significantly decreased (i.e., X% in this study under that condition) on the affected side compared to the healthy one. "

Response 9: Thank you for your valuable feedback. I completely agree with your suggestion regarding adding quantitative information to the conclusion section. I have revised as described below. “The velocity of CHL was significantly decreased (11.2% in mean velocity and 24.8% in maximum velocity in this study under that condition) on the affected side.”

Point 10: In addition, I understand that it is the knowledge in the medicine that US does not present a high SNR, but has many advantages compared to other modalities. One of the advantages of using US is the lower cost and no use of radiation (e.g., compared to CT studies or guided procedures). It might be interesting to discuss this in the introduction since the readers of SENSORS might not know such advantages. 

Response 10: Thank you for your feedback. I appreciate your comments on the benefits of US compared to other modalities, such as CT studies or guided procedures. I have mentioned the benefits of using US in the section on TFCC injury in the introduction as described below. “Magnetic resonance imaging (MRI) and computed tomographic arthrography (CTA) are commonly used to diagnose TFCC injury, but these imaging modalities are costly and radiation-exposed, whereas US has the advantage of low cost and no radiation exposure.”

Reviewer 2 Report

"Yang et al. grossly examined the insertion of the CHL in 26 fresh-31 frozen cadavers." 

The distal insertion of the CHL presents a complex spatial architecture that should be better described in the text to define its pivotal role in modulating the humeral head rotations. The CHL is in a histological continuum with the rotator cable of the shoulder and presents three main bands. The anterior band is attached to the lesser tubercle of the humeral head, the intermediate band is attached to the anterior corner of the greater tubercle, and the posterior band is attached to the posterior corner of the lesser tubercle of the humeral head. In this sense, the CHL "control" the glenohumeral joint rotations through a wide fan distribution of its fibers distally. Moreover, as stated by the authors in the limitations of the study, the abnormal motion of the CHL can also be partially related to the rotator cuff tear considering the direct continuation of the CHL with the rotator cable of the shoulder. Lastly, the aforementioned anatomical details of the CHL-rotator cable complex are pivotal to better clarify the clinical outcomes induced by the ultrasound-guided interventions to the CHL (e.g., injections, perforations, etc). 

"Arai et al. examined the structure between the CHL and subscapularis histo-anatomically in nineteen cadavers and discussed the function of the ligament."

The authors should at least mention the other histological elements present in between the coracoid process and subscapularis muscle because they can be involved in the clinical scenario of the "frozen shoulder" as well as the CHL. In this sense, the subcoracoid fat pad, the subcoracoid bursa, and the subscapular recess of the glenohumeral joint are the main anatomical structures at this level. Adhesions in between the aforementioned histological layers are frequently identified during arthroscopy and treated accordingly. Of note, adhesions at this level are directly responsible for a reduced velocity of the CHL during shoulder rotations in the clinical scenario of shoulder contracture (also in patients with a normal thickness of the CHL!). For the subcoracoid space anatomy and pathology, please refer to J Ultrasound Med. 2022 Sep;41(9):2149-2155.]   

Positioning the probe in a transverse plane (Figure 1), the CHL ligament can be visualized but we can't define it as a "complete" long-axis view of the ligament with requires an oblique orientation of the probe. In this sense, I suggest the authors revise the legends of the figures accordingly.  

For figure 1 I suggest adding more photos of the different angles of rotations of the shoulder filling the entire column of the manuscript. In this sense, the figure is more complete and the editing of the manuscript is more organized. 

For figure 2 I suggest: "the triangular-in-shape hyperechoic structure located between the subscapularis tendon, the coracoid process, and the deltoid muscle was defined as the CHL in the present study".  

Figure 6. The ultrasound images in (a) and (b) are very similar almost the same. They should be in internal and external rotations. Please, check them. 

I suggest the authors add an ultrasound video (supplementary material) to clearly show the dynamic maneuver in order to allow readers a satisfactory reproducibility of the test.

Author Response

Point 1: "Yang et al. grossly examined the insertion of the CHL in 26 fresh-31 frozen cadavers." The distal insertion of the CHL presents a complex spatial architecture that should be better described in the text to define its pivotal role in modulating the humeral head rotations. The CHL is in a histological continuum with the rotator cable of the shoulder and presents three main bands. The anterior band is attached to the lesser tubercle of the humeral head, the intermediate band is attached to the anterior corner of the greater tubercle, and the posterior band is attached to the posterior corner of the lesser tubercle of the humeral head. In this sense, the CHL "control" the glenohumeral joint rotations through a wide fan distribution of its fibers distally. Moreover, as stated by the authors in the limitations of the study, the abnormal motion of the CHL can also be partially related to the rotator cuff tear considering the direct continuation of the CHL with the rotator cable of the shoulder. Lastly, the aforementioned anatomical details of the CHL-rotator cable complex are pivotal to better clarify the clinical outcomes induced by the ultrasound-guided interventions to the CHL (e.g., injections, perforations, etc). 

Response 1: Thank you for your feedback. The following content and references regarding the role of the CHL in the glenohumeral joint have been added. “Boardman et al. considered the CHL as an important capsuloligamentous structure of the glenohumeral joint [4]. The CHL acts in combination with the rotator cuff muscles, superior glenohumeral ligament, and capsule to contribute to the stability of the glenohumeral joint. In particular, it functions to prevent inferior subluxation of the humeral head and plays an important role in stabilizing the glenohumeral joint in the upright position [5].”

Point 2: "Arai et al. examined the structure between the CHL and subscapularis histo-anatomically in nineteen cadavers and discussed the function of the ligament." The authors should at least mention the other histological elements present in between the coracoid process and subscapularis muscle because they can be involved in the clinical scenario of the "frozen shoulder" as well as the CHL. In this sense, the subcoracoid fat pad, the subcoracoid bursa, and the subscapular recess of the glenohumeral joint are the main anatomical structures at this level. Adhesions in between the aforementioned histological layers are frequently identified during arthroscopy and treated accordingly. Of note, adhesions at this level are directly responsible for a reduced velocity of the CHL during shoulder rotations in the clinical scenario of shoulder contracture (also in patients with a normal thickness of the CHL!). For the subcoracoid space anatomy and pathology, please refer to J Ultrasound Med. 2022 Sep;41(9):2149-2155.]

Response 2: Thank you for bringing this to my attention. The following content and references regarding the other histological elements present in between the coracoid process and subscapularis muscle have been added. “Other histologic elements present between the coracoid process and the subscapularis muscle include the subcoracoid fat pad, the subcoracoid bursa, and the subscapular recess of the glenohumeral joint [2].”

Point 3: Positioning the probe in a transverse plane (Figure 1), the CHL ligament can be visualized but we can't define it as a "complete" long-axis view of the ligament with requires an oblique orientation of the probe. In this sense, I suggest the authors revise the legends of the figures accordingly. For figure 1 I suggest adding more photos of the different angles of rotations of the shoulder filling the entire column of the manuscript. In this sense, the figure is more complete and the editing of the manuscript is more organized.

Response 3: Thank you for your feedback. We appreciate your suggestion to revise the legends of the figures to reflect the limitations of visualizing the CHL ligament in a transverse plane. We have changed photos from the front and added photos from above.

Point 4: For figure 2 I suggest: "the triangular-in-shape hyperechoic structure located between the subscapularis tendon, the coracoid process, and the deltoid muscle was defined as the CHL in the present study". 

Response 4: Thank you for your valuable input. We have made the suggested change to the manuscript regarding Figure 2, as follows: “the triangular-in-shape hyperechoic structure located between the subscapularis tendon, the coracoid process, and the deltoid muscle was defined as the CHL in the present study”.

Point 5: Figure 6. The ultrasound images in (a) and (b) are very similar almost the same. They should be in internal and external rotations. Please, check them. 

Response 5: We are very sorry that the images are not clear. The direction of the arrow is different in (a) and (b).

Point 6: I suggest the authors add an ultrasound video (supplementary material) to clearly show the dynamic maneuver in order to allow readers a satisfactory reproducibility of the test.

Response 6: We appreciate your suggestion and agree that providing an ultrasound video could enhance the reproducibility of our study. We have included a supplementary material video demonstrating the dynamic maneuver in order to provide readers with a better understanding of the procedure. Thank you for your valuable feedback.